# In Silico Protein Folding Prediction of COVID-19 Mutations and Variants

**DOI:** 10.3390/biom12111665

**Published:** 2022-11-10

**Authors:** Sumana Bhowmick, Tim Jing, Wei Wang, Elena Y. Zhang, Frank Zhang, Yanmin Yang

**Affiliations:** Department of Neurology and Neurological Sciences, School of Medicine, Stanford University, 1201 Welch Road, MSLS, P259, Stanford, CA 94305, USA

**Keywords:** trRosetta, AlphaFold, SARS-CoV-2, receptor-binding motif, angiotensin-converting enzyme 2 receptor, HADDOCK

## Abstract

With its fast-paced mutagenesis, the SARS-CoV-2 Omicron variant has threatened many societies worldwide. Strategies for predicting mutagenesis such as the computational prediction of SARS-CoV-2 structural diversity and its interaction with the human receptor will greatly benefit our understanding of the virus and help develop therapeutics against it. We aim to use protein structure prediction algorithms along with molecular docking to study the effects of various mutations in the Receptor Binding Domain (RBD) of the SARS-CoV-2 and its key interactions with the angiotensin-converting enzyme 2 (ACE-2) receptor. The RBD structures of the naturally occurring variants of SARS-CoV-2 were generated from the WUHAN-Hu-1 using the trRosetta algorithm. Docking (HADDOCK) and binding analysis (PRODIGY) between the predicted RBD sequences and ACE-2 highlighted key interactions at the Receptor-Binding Motif (RBM). Further mutagenesis at conserved residues in the Original, Delta, and Omicron variants (P499S and T500R) demonstrated stronger binding and interactions with the ACE-2 receptor. The predicted T500R mutation underwent some preliminary tests in vitro for its binding and transmissibility in cells; the results correlate with the in-silico analysis. In summary, we suggest conserved residues P499 and T500 as potential mutation sites that could increase the binding affinity and yet do not exist in nature. This work demonstrates the use of the trRosetta algorithm to predict protein structure and future mutations at the RBM of SARS-CoV-2, followed by experimental testing for further efficacy verification. It is important to understand the protein structure and folding to help develop potential therapeutics.

## 1. Introduction

The determination of protein 3D structure is crucial to understand its biological function. However, it is time-consuming and expensive. For several decades, the prediction of protein structure directly from sequence information has been an unachievable dream. The advancements in different computational methods to predict protein tertiary structures have now allowed one to study folds and local motifs, molecular folding, evolution, and structure/function relationships in proteins. The BigNet model was the first proposed application of a back propagation neural network (BPNN) to predict the protein 2D and 3D structures [1]. In 1994, Burkhard Rost et al. used evolutionary information as multiple sequence alignments in neural networks as an input which significantly increased the accuracy of the protein prediction [2]. K. T. Simons et al. used simulated annealing and Bayesian scoring function to generate tertiary protein structure based on fragments of unrelated proteins [3]. To date, various methods based on machine learning programs, such as feedforward neural networks, two input neurons, radial basis function neural networks, two neural networks (SPINE-2D), SVM-based contract predictors [4,5,6,7,8], and deep learning programs, such as bidirectional recurrent neural network, residue-residue coevolution, a balanced network deconvolution (BND), and deep neural network [9,10,11,12], have been proposed to find protein tertiary structure. However, proteins usually function through interactions with other proteins or molecules and predicting protein interactions has been even more challenging. For each structure prediction, many short simulations starting from different random sequences are carried out to generate an ensemble of decoy structures that have both favorable local interactions and protein-like global properties [13]. The substantial progress of blind protein prediction using protein ranking refinement and prediction of multimeric complexes has been acknowledged in the Critical Assessment of Structure Prediction (CASP) competition [14,15,16,17]. Different techniques have been developed, and benchmarks have been drawn to elucidate the advantages and disadvantages of different methods. For instance, Google’s DeepMind developed AlphaFold, a highly ranking method [12]. Another example is the Zhang server, built on iTasser and QUARK [14]. Protein structure prediction using AlphaFold and Rosetta algorithms have been widely used methods and perhaps the most successful methods for de novo protein structure prediction since its inception [12,13,18]. Additionally, several databases and coevolution have been used to predict these interactions [19,20,21].

DeepMind developed AlphaFold1, which ranked at the top in CASP13 [22]. Inspired by its successes, Baker lab proposed that limitations in AlphaFold1 could be further improved by using inter-residue orientations as well as distance measurements [23]. “trRosetta uses both residue-residue distances and orientations, which gives richer information on the structure compared to distances only,” explained Baker [24]. The components of the method trRosetta include (1) a deep residual-convolutional network which takes an MSA as the input and outputs information on the relative distances and orientations of all residue pairs in the protein and (2) a fast Rosetta model building protocol based on restrained minimization with distance and orientation restraints derived from the network outputs [25]. trRosetta helps predict inter-residue orientations and distances from co-evolutionary data by applying deep knowledge, which significantly improves protein structure prediction [23] from amino acid sequences. One of the key features of trRosetta is that it originally had de novo modeling. However, later automated template detection by HHsearch was included to improve the accuracy. It has been reported that sequence design using trRosetta has a remarkable ability to capture properties of the energy landscape and consider alternative states that can reduce the occupancy of the desired target structure [24,26]. As per Du et al., since its release in 2020, the standalone package has been downloaded by >2000 registered users [23]. The trRosetta models were shown to fit well with the cryo-electron microscopy experimental data [27,28]. They have been intensively used to understand the structure and function of lipid transporters [29], protein function deficiency [30], the *Magnaporthe oryzae* secretome [31], the structural characterization of S59L for efficient treatment for ALS and FTD diseases [32], 3D modeling of antibody domains [33], predicting vaccine construct [34], and more. The results of these studies have encouraged researchers and beyond. However, in CASP14, DeepMind’s AlphaFold2 replaced almost all components of AlphaFold1, solving the single-chain-based protein structure prediction problem and creating a milestone [35].

The COVID-19 pandemic has become an international event leading to international devastation, closing, affecting all aspects of normal life. However, the introduction of vaccines in the first half of 2021 caused an evolutionary shift toward immune-escaping variants. The most notable, Omicron BA1, was discovered in South Africa [1,2]. Thus, it is imperative to understand the reasons why certain mutations were able to cause such significant viral changes in pathogenicity, virulence, and transmissibility [3]. Various studies have proposed or undergone in vitro cultivation and examination of the RNA and protein sequences of the viruses, and their results further our understanding of COVID-19 [4]. However, large-scale prophylactic work has not been conducted as the scope of such laboratory experiments is constrained by the time required to culture and mutate individual pathogens when attempting to evaluate for different mutations. However, experimental structures of many proteins are still not available to date. Thus, recently, computational methods have been gaining in popularity compared to experimental methods. According to previous studies on vaccine efficacy, the Receptor-Binding Domain (RBD), a part of the S1 subunit in the spike (S) (Appendix A) protein of the SARS-CoV-2 virus, interacts directly with the angiotensin-converting enzyme 2 (ACE-2) (EC:3.4.17.23) receptor within the susceptible host. The S glycoprotein facilitates the introduction of viral material into the host cell [36,37,38]. The domain is located between residues 319–541 of the spike protein. Thus, any potential mutations within the RBD or the S protein may present significantly stronger mechanisms of cell infiltration, increasing the ability of the virus to infect human cells [39]. A thorough examination of the RBD protein structure and potential mutation residues can allow for predictions and, thus, prevention against even more deleterious variants that may arise in the future, saving lives. A novel and promising solution is the implementation of de novo protein folding algorithms to generate a large database of variants, followed by protein–protein interaction analysis to determine the protein function [9], diagnosis [10], and phylogenetic analysis [11].

In this study, we use de novo or ab initio prediction of the 3D structures of the receptor-binding sequences of COVID-19 using trRosetta algorithm, a separate algorithm from AlphaFold v2. Subsequently, refinement of predicted complexes is conducted using HADDOCK, an information-driven flexible docking approach; and PRODIGY, a binding affinity descriptor based only on structural properties of a protein–protein complex between the RBD mutations and the ACE-2 receptor. We predicted the 3D structures of different naturally occurring variants of COVID-19 and compared them with the existing structure from an experimental PDB structure. This provided insight into variations of the RBD and its interactions with the ACE-2 receptor in relation to infiltration and infectivity, leading us to the residues of the Receptor-Binding Motif (RBM). We further predicted potential mutations at specific conserved RBM residues of different naturally occurring variants using trRosetta. We also predicted the mutations’ interactions with the ACE-2 receptor using HADDOCK and PRODIGY. Our study revealed certain mutations yet to be in existence with the potential to have higher binding affinity to the ACE-2 receptor. Additionally, the efficacy of the proposed mutation was further assessed using experimental testing. This study introduces a pipeline methodology that can be used in future studies of COVID-19 and as a guide for strategies to combat new/old diseases.

## 2. Materials and Methods

### 2.1. Sequence Identification

The spike protein sequence for the Original Wuhan strain of COVID-19 (Gene ID: 43740568, Sequence: NC_045512.2) was retrieved from the NCBI database (https://www.ncbi.nlm.nih.gov/ accessed on 23 June 2021) in FASTA format. In addition, we generated the sequences of the Alpha, Beta, Gamma, Delta, and Omicron variants by manually editing the original Wuhan strain with the RBD residue mutations provided by the CDC (https://www.cdc.gov/coronavirus/2019-ncov/variants/variant-classifications.html, accessed on 23 June 2021) (Appendix A).

### 2.2. De Novo Protein Folding and Structural Analysis

The SARS-CoV-2 Receptor-Binding Domain (RBD), residues 319–541, of each variant, were selected for de novo protein folding. We initially used both transform-restrained Rosetta (trRosetta) (https://yanglab.nankai.edu.cn/trRosetta/ accessed on 23 June 2021) and AlphaFold v2.1.0 de novo protein folding algorithms, respectively, for the sequences of the six variants (Original, Alpha, Beta, Gamma, Delta, and Omicron) to construct a 3D computer model of the protein. The multi-sequence alignment (MSA) inputs were conducted using the HH-suite3 script (https://github.com/soedinglab/hh-suite, accessed on 23 June 2021) with operational instructions available on an open-source GitHub repository [13]. The HH-suite3 script uses the UniRef30 database (https://colabfold.mmseqs.com/ accessed on 23 June 2021) of MSAs to search with the target sequence using HHblits with all subsequent instructions detailed in the GitHub ReadMe [14]. The multi-sequence alignment features were then uploaded into the trRosetta online portal, where MSA files can be uploaded to generate the final PDB model with default parameters [15].

The AlphaFold algorithm AlphaFold v2.1.0 has been released as open source by Deep Mind, with a version available for use on a Colab notebook [12]. The sequences were directly run as per the directions on the Colab notebook (https://colab.research.google.com/github/deepmind/alphafold/blob/main/notebooks/AlphaFold.ipynb, accessed on 23 June 2021) with enough computational power, such as through utilizing Colab Pro’s access to faster GPUs and TPUs. The AlphaFold algorithm produces Protein Data Bank (PDB) files, which is a way to encode a 3D computer protein model.

### 2.3. Visualizing and Preliminary Analysis

The models generated from trRosetta and AlphaFold v2.1.0 for the naturally occurring variants were then superimposed and aligned using the PyMol software (https://pymol.org/2/ accessed on 23 June 2021) [16] against a laboratory model of the RBD–ACE-2 complex (PDB: 6M0J) retrieved from the Protein Data Bank (https://www.rcsb.org, accessed on 23 June 2021). We utilized the PyMol super command, which super aligns two protein selections. Super does a sequence-independent structure-based dynamic programming alignment (unlike the align command) followed by a series of refinement cycles intended to improve the fit by eliminating pairing with high relative variability [17]. The super command is more reliable than the align command for proteins with low sequence similarity, generating a numerical Root-Mean-Square-Deviation (RMSD) value for quantification.

### 2.4. Docking Analysis

The receptor-binding domains were then computationally docked with the ACE-2 receptor using High Ambiguity Driven Biomolecular DOCKing (HADDOCK 2.4) [17] (https://wenmr.science.uu.nl/haddock2.4/ accessed on 23 June 2021). All the sequences were divided into two sets; Molecule 1 was the RBDs produced from trRosetta, and Molecule 2 was the ACE-2 receptor from PDB: 6M0J. HADDOCK 2.4, an experimentally based docking program, was used to dock Molecule 1 to Molecule 2 uploaded respectively with a default setting. To launch the docking process, we selected the active residues for Molecule 1, roughly 151-186 aa, which was closest to the ACE-2 receptor (residues 1-48 aa) of PDB: 6M0J’s interaction interface that is corroborated by literature [19]. Furthermore, the PRODIGY (protein binding energy prediction) webserver (https://wenmr.science.uu.nl/prodigy/ accessed on 23 June 2021) was used to calculate the binding affinity either as Gibbs free energy (ΔG, kcal/mol) or as a dissociation constant (Kd, M) at 25 °C between the RBD’s and the ACE-2 receptor. Molecule 1 and Molecule 2 were labeled as different chains and merged into a single file using PyMol to upload with a default setting [20].

### 2.5. In Vitro Test

All chemicals are reagent grade and were obtained from Millipore-Sigma-Aldrich Co., St. Louis, MO, USA and all tissue culture reagents were from Life Technologies (Invitrogen TM) Framingham, MA, USA.

SARS-CoV-2 Spike protein (Sp) expression constructs: Spike protein (Sp) constructs were used in the experiments: mRFP-Spike Protein (Sp) Original (mRFP-Sp WT, EX-NV219-M55) and Kappa (*B.1.617.1*, mRFP-Sp Kappa, EX-NV248-M55) were purchased from Genecopoeia, Inc. Rockville, MD, U.S.A. Kappa includes the following eight mutations: K19R, G142D, E154K, L452R, E484Q, D614G, P681R, and Q1031H. Human ACE-2 conjugated mono-GFP (ACE-2-GFP, EX-U1285-M98) was also acquired from Genecopoeia, Inc.
(1)Based on the kappa plasmid, the delta (*B.1.617.2*, mRFP-Sp Delta) was generated in the lab with the following nine mutations: T19R, 156&157del, R158G, L452R, T478K, D614G, P681R, and Q1031H.(2)As predicted, Threonine (Thr) at residue 500 of the delta variant was mutated to Arginine (Arg) to obtain the T500R mutations (mRFP-Sp Delta-T).Antibodies: Primary antibodies used were SARS-COV-2 Spike Protein (E7B3E) (Cell Signaling Technology, Inc. Danvers, MA, U.S.A. #63847, rabbit monoclonal, 1:2000 for WB, 1:200 for IF). SARS-COV-2 Spike Protein (2B3E5) (Cell Signaling Technology, Inc. #52342S, mouse monoclonal, 1:2000 for WB, 1:200 for IF). SARS-CoV-2 Spike RBD (R&D Systems, Inc. Minneapolis, MN, U.S.A. AB#105401, mouse monoclonal, 1:2000 for WB, 1:200 for IF). Beta-actin (13E5) (Cell Signaling Technology, Inc. #4970, rabbit monoclonal, 1:2000 for WB, 1:200 for IF). Beta-actin (C4) (Santa Cruz Bio, Santa Cruz, CA, USA, SC-47778, mouse monoclonal, 1:1000 for WB, 1:100 for IF). Secondary antibodies for immunofluorescence were Alexa Fluora^®^ 488- and Alexa Fluor^®^ 594 conjugated secondary antibodies (Jackson ImmunoResearch, West Grove, PA, USA) and IRDye 800CW Goat anti-Rabbit (Li Cor Biosciences, Lincoln, NE, USA, 926-32211, 1:20,000) and IRDye 680LT Goat anti-Mouse IgG (Li Cor Biosciences, 926-68020, 1:20,000) for Western blotting.Cells lines and culture: HEK 293T cell line (CRL-11268^TM^, ATCC) and COS7 cells (CRL-1651^TM^, ATCC) were cultured in Dulbecco’s Modified Eagle’s Medium (DMEM) with 10% FBS and incubated in a humidified incubator at 5% CO_2_ and 37 °C.Soluble Spike proteins preparation from cells (Freeze & Thaw lysis): 293T cells were set up in p100 dishes and transfected with 8 μg of different Sp plasmids with Lipofectamine 3000 (Thermo Fisher Scientific, Waltham, MA, USA) for 2 days. Cells were then scraped and transferred to 1.5 mL microfuge tube with 1 mL ice-cold PBS. After centrifuge, the pellet was resuspended in 500 μL lysis buffer (600 mM KCl, 20 M Tric-Cl (pH 7.8), and 20% Glycerol) with protease inhibitor (Roche Applied Sciences, Penzberg, Germany). The tube was sealed and frozen using liquid nitrogen, then thawed on ice. After three rounds of the “Freeze–Thaw” process, 250U of Benzonase was added to digest DNA for 10 min at RT (or alternatively via sonication). After a quick vortex, the samples were centrifuged with maximum speed for 30 min in a cold room, and 100 μL per tube of supernatants was aliquoted and saved as ligand samples.Ligand bound assay and Western blot analysis: COS7 cells were set up in p60 dishes. After overnight plating, the total amount of Sp proteins (around 300 ng) was added to the cells. Then, 2 h after incubation, the cell dishes were washed twice with PBS for 5 min to remove unbound ligands. A total of 125 μL SDS sample loading buffer was used to lysis cells and collected into the 1.5 mL micro-centrifuge tubes. The cell lysates in the tube were heated at 100 °C for 7 min and centrifuged for 5 min. The standard purified His-Sp (R & D System) was used as control for quantification. One-third of the total lysates were loaded to sodium dodecyl sulfate-polyacrylamide gel electrophoresis (SDS-PAGE) gel and transferred to a PVDF membrane (Immobilon-FL membrane, 0.45 μm, IPFL00010). The membranes were blocked in Intercept Blocking buffer (Li Cor Biosciences, 927-60001) for 30 min at room temperature and then incubated overnight at 4 °C with primary antibodies. The next day, membranes were incubated at room temperature for 1 h with secondary antibodies. The fluorescent signal was assayed using an Odyssey Infrared Imaging system (Li Cor Biosciences).Immunofluorescence: COS 7 Cells cultured on coverslips after spike protein adding were fixed in 4% paraformaldehyde for 12 min at room temperature (RT), then permeabilized TBS with 0.1% Triton X-100 for 5 min. After blocking with 5% BSA/TBS for 30 min at RT, the coverslips were incubated with either primary individual antibodies or combined antibodies from different species in at cold room overnight, followed by incubation of primary antibodies overnight, secondary antibodies for 1 h and DAPI-Fluoromount-G^®^ (0100-20; South-ern Biotech, Birmingham, AL, USA) stain. Images were analyzed using a fluorescence microscope (Nikon Image system).Ligand-receptor endosomal assay: COS7 cells were cultured on coverslips overnight and added with ligand (mRFP-Sp) for 2 h, then cells on a coverslip were quickly treated with 0.025% Trypsin for 1 min after being washed with PBS. Next, cells were stained following the immunofluorescence procedures. Heatmap was considered for the intensity comparison (LUT).Cell in-fusion assay: 293T cells were set up in p60 dishes and individually transfected with 3 µg of different mRFP-Sp plasmids (Sp Delta and Sp Delta T500R), mRFP, eGFP, and ACE-2-GFP with Lipofectamine 3000 for overnight. All transfected cells were trypsinized and resuspended in a 4 mL culture medium. Different combinations were set (eGFP + mRFP, eGFP + mRFP-Sps, ACE-2-eGFP + mRFP, and ACE-2-eGFP + mRFP-Sps), an example like: 0.5 mL of ACE-2-eGFP transfected 293T cells was mixed with 0.5 mL of 293T cells with mRFP-Sp delta, in 15mL falcon tubes for 15 min at 37 °C cell incubator, then diluted in the 3mL culture medium and settled down on the coverslips for live imaging and were monitored for 4 h, 8 h and overnight to check cell in fusion statues. In-fusion percentages of ACE-2-eGFP were compared among different spike protein variants or mutations.Statistical Analysis: The data were analyzed by Student’s *t*-test or two-tailed *t*-tests (two-way ANOVA) with the Prism 8 (GraphPad) or Excel (Microsoft) software. *p*-values < 0.05 were considered statistically significant.

## 3. Results

Previous bioinformatic analyses have shown that the spike protein RBD generates proteins that are essential for host binding and prone to new mutations [40,41]. To investigate and compare the RBD, we initially used the amino acid sequences of the Original variant obtained from NCBI and then manually edited the residues for the other five naturally occurring variants, namely the Alpha variant, Beta variant, Gamma variant, Delta variant, and Omicron variant (Appendix A). These sequences were then processed using trRosetta suite and AlphaFold v2.1.0 individually. Although the experimentally solved structures are available on public databases, our aim was to evaluate the algorithm so that it can regularly predict protein structures even in cases when no similar structure is known/available. The predicted models generated by trRosetta and AlphaFold v2 were then superimposed against the laboratory derived PDB: 6M0J through the “super” command of PyMol (Figure 1). We observed that for most pairs of the trRosetta-generated models and PDB: 6M0J residues, the RMSD ranges from 0.69 to 0.77 Å, and AlphaFold-generated models give RMSD ranges from 0.83 to 1.6 Å. Lower RMSD indicates a good alignment of the RBDs. Thus, we considered trRosetta-generated models for further analysis (Table 1). Qualitative structural changes observed through superimposing the trRosetta-generated naturally occurring variants in Figure 1A-a suggest that the mutations were focused on the interaction interface located on the bottom section of the RBD (Figure 1A-b,A-c). The Receptor-Binding Motif (RBM) is the main functional motif in the RBD, and upon highlighting the residues (Figure 1B-a) of 6M0J, we observe structural changes in comparison to the Delta and Omicron sequences (Figure 1B-b,B-c).

The trRosetta-generated structures of RBD were then docked with the ACE-2 receptor from 6M0J using the HADDOCK server. The HADDOCK program generated several complexes and clustered them. The cluster with the highest HADDOCK score and lowest electrostatic energy was selected. The best complex was chosen based on cluster size, HADDOCK score, and electrostatic energy. The HADDOCK generates water-refined interaction conformations of both interacting molecules and groups similar conformations in clusters. The superimposed structures of all the complex structures in one cluster were generated along with the plots of HADDOCK scores, Cluster size, RMSD, Van der Waals energy, electrostatic energy, desolvation energy, restraint violation energy, buried surface area, and Z-score. The best possible interaction complex for all receptors with the best HADDOCK score was then selected for refinement (Appendix A). Considering the fact that both the Original variant sequence and the crystal structure of SARS-CoV-2 spike RBD in 6M0J was from the Wuhan strain of COVID-19, the HADDOCK scores from the RBD-ACE2 complex of the Original variant and the 6M0J were taken as thresholds. The docked complexes were further analyzed in PRODIGY for the number and types of interactions between the RBD and the ACE-2 receptor. PRODIGY generates a binding affinity value (ΔG) and a total number of interactions between both molecules. Additionally, the information on the total number of charged–charged interactions, charged–polar, charged–apolar, polar–polar, polar–apolar, and apolar–apolar interactions was generated in the analysis provided in Appendix A.

Previous literature has established that the spike protein of SARS-CoV-2 binds to the ACE2 receptor via its receptor-binding domain (RBD). The cartoon representation of the refined structure and interaction complexes generated by PRODIGY was then visualized on PyMol (Figure 2). We observe that the binding affinity (ΔG) of the Alpha and Omicron variants are the lowest (−14.5 kcal/mol and −14.2 kcal/mol, respectively). The higher binding interaction with the ACE2 receptor might reflect a greater ability for potential immune evasion risk and high transmissibility [42] as we have observed with the Omicron variant. We also observe the presence of hydrogen bonding between G496 of the RBM and K671 of the ACE-2 receptor of the Original variant. In addition, T500 changes for the Alpha, Beta, and Gamma variants, where we can also see it bind to P499 of the RBM (Figure 2D–F). However, the binding of G496 is consistent between the Delta and Omicron variants, even with the natural mutation of G496S in Omicron (Figure 2G,H).

Considering the variations of interactions observed between the residues of the RBM of the naturally occurring variants binding to K671 of the ACE-2 receptor, we generated new variants with mutations on the corresponding amino acids on the RBM of the naturally occurring variants and grouped them as Variant 1 to Variant 11 (Table 2). To understand if the change in the interaction between K671 and P499/T500 affects the binding affinity, Variant 1 was generated with all the mutations found in the Alpha and Delta variants. Then, we generated various known point mutations in the Original variant, such as N501Y (Variant 2) and S494P (Variant 3). Since S494P is absent in the Delta and Omicron variants (Appendix A), we generated those mutations to see the alterations in their interactions (Variants 4 and 5). We then generated mutations at P499 and T500, as these were part of conserved residues in the naturally occurring variants. Considering the Mutagenetix database, http://mutagenetix.utsouthwestern.edu accessed on 23 June 2021, we selected the most possible incidental mutation of Proline to Serine (Variant 6, 7, and 8) for P499 and the least possible mutation of Threonine to Arginine (Variant 9, 10, and 11) for T500.

All the “artificial” variants underwent the same aforementioned methodology to generate a PDB structure using trRosetta with RMSD ranging from 0.59 to 0.81 Å, which indicated a good alignment of the RBDs (Appendix A) and was further analyzed using HADDOCK and PRODIGY.

The trRosetta-generated structures of RBDs of the artificially generated variants were then docked with the ACE-2 receptor from PDB: 6M0J using the HADDOCK server. Variants 2, 3, 6, 7, and 8 showed lower HADDOCK scores compared to the naturally occurring variants, suggesting a stronger interaction between their RBDs and the ACE-2 receptor (Appendix A). Taking into account that Variant 2 [43] and Variant 3 [43] are already recognized and have been reported to enhance the affinity of RBDs to the ACE-2 receptor, we considered looking through the other mutations that currently do not exist, such as Variant 6 (−123.5 ± 1.2), Variant 7 (−111.8 ± 2.9), and Variant 8 (−127.1 ± 3.1). These were further analyzed in PRODIGY for the number and types of interactions (Appendix A).

The cartoon representation of the refined structure and interaction complexes generated by PRODIGY is provided in Figure 3B,D. The P499S mutation in Delta showed new interactions of S499 with T41 of the ACE-2 receptor (H-bond length of 3.2 Å), G498 with A38 (H-bond length of 2.1 Å), and that G496 bonds with L353 (H-bond length of 2.5 Å) instead of K671 (H-bond length of 3.2 Å) (Figure 3B). The P499S mutation in Omicron shows interactions of T500 and S496 with K671 of the ACE-2 receptor (H-bond length of 3.5 Å and 3.2 Å, respectively), which was initially absent in the naturally occurring Omicron variant. The mutation also changes the interaction of Y501 with G644 to Y501 with T642 (Figure 3D). The interaction was further studied through PRODIGY, where the presence of P499S in Delta and Omicron showed binding affinities of −12.2 kcal/mol and −14.5 kcal/mol, respectively, in comparison with Delta to ACE-2 (−13.8 kcal/mol) and Omicron to ACE-2 (−14.2 kcal/mol) (Figure 3). Additionally, both P499S mutations in Delta and Omicron show increased electrostatic energy in comparison to the mutation in the Original variant, which might be responsible for the higher binding affinity. To further understand the interaction of T500 with K671, we created T500R mutations in the Original, Delta, and Omicron variants. We observed a major difference in the binding affinity between the naturally occurring variants and the mutated variants. This was reflected in the cartoon representation, which suggests a complete loss of the H-bond of K671 in Delta and of G644 in Omicron. However, both mutations still show a decent binding affinity compared to other variants (Figure 3). Additionally, variations in electrostatic energy and Van der Waals energy were clearly observed, which might be responsible for the alterations in binding affinity.

To determine the efficacy of whether the predicted T500R mutation changes the cell binding affinity in vitro, we carried out some preliminary tests for its binding to the spike protein and infiltration using a cell-cell fusion assay. Initially, plasmids of the Delta variant and the T500R Delta variant were made in conjugation with mRFP as described in Materials and Methods (Appendix A). To prepare the ligand in natural statue, soluble lysates were prepared via the “Freeze–Thaw” method from transfected 293T cells (Appendix A). After filtering through the low retention 0.25 µm filter, the cell lysates containing spike proteins were added into pre-cultured COS7 cells either on dish for Western blot or coverslip for staining. Compared with mRFP, the mRFP-Sp was detected on cells, with a majority surrounding the membrane, and the 100 ng/mL saturation scale was reached (Appendix A). To load the same amount of spike proteins, the lysates from the Delta variant and its mutation were quantified via Western blot with standard purified spike protein (purchased from company) (Figure 4A-a). Then, the cell-bound assay was performed to dissect the difference between the Delta variant and its mutation (T500R). Figure 4A-b surprisingly shows that the T500R mutation enhanced the cell bound to 30% (Figure 4C) after statistical analysis and was further confirmed with immunofluorescence (Figure 4B).

SARS-CoV-2 uses clathrin-mediated endocytosis to gain access to cells, and thus, endosomes tend to form inside cells once ligands associate with receptors [44]. More endosome formation was observed for the Delta T500R mutation under heat map (LUT) presentation. To investigate whether the increased association of T500R on cells was through ACE-2, ACE-2-eGFP plasmid was transfected and confirmed correct expression pattern via antibody (Appendix A). Next, ACE-2-eGFP and mRFP-Sps (Delta and its T500R mutation) were transfected for cell in-fusion assay (Appendix A). This was to monitor the binding activity of the spike proteins by using SARS-CoV-2-S-transfected 293T as effector cells and ACE-2-expressing 293T as target cells [45]. The cells in the fusion phenome were only observed in the combination of ACE-2-eGFP and mRFP-Sps instead of others (Appendix A). Compared with Delta, the T500R mutation with ACE-2-eGFP exhibited a high capacity for membrane fusion and was efficient in mediating virus fusion and entry into target cells.

The in vitro data suggest that T500R enhances the spike protein’s efficacy for fusion via the ACE-2 receptor’s binding affinity and correlates with the in-silico results. These results indicate that our algorithm and simulations agree with the in vitro results, and thus mutations at P499 or T500 that equally increase the RBM’s binding affinity to ACE-2 could still be a potential target.

## 4. Discussion

The widely applied and most popular web-based platform trRosetta suite has resolved some of the greatest challenges in protein folding, presenting a new era of opportunities. It highlights the combination of Rosetta-based optimization with predicted information and additional components of the Rosetta energy function, thus generating protein models. This allows researchers to access de novo protein structures in future research, as protein folding remains a challenging field among researchers. Probing this technology and its myriad uses will similarly yield numerous applications that can further our fundamental understanding of life. We here evaluate and demonstrate an accurate and fully automated pipeline to predict protein folding and its key interactions with receptors. This specific approach of using trRosetta algorithm was intriguing to us because it was an available web-based platform and presented a scalable, efficient method for future researchers to conduct preliminary analysis into new mutations and variants that may not yet exist under natural conditions. Additionally, several studies have been reported on protein/peptide prediction of/against novel coronavirus SARS-COV-2 using trRosetta modeling suite [23,46,47,48]. Instead of time-consuming laboratory work to alter the genetic sequences of pathogens, large-scale computer generation of proteins could provide an avenue to narrow down mutations to study, discern patterns that can only be seen with large quantities of data and identify new therapeutics.

Previous studies conducted on COVID-19 pathogenesis suggest that its spike protein is critical in both ACE-2 receptor binding and viral fusion into the host cell membrane [49]. Indeed, the spike protein has also been the primary immunogen target for the currently available COVID-19 vaccines [50,51]. Until now, a countless number of mutations (e.g., T19R, A67V, D80A, D80G, T95I, G142D, Y145D, R158G, L212I, D215G, A222V, N234Q, W258L, N331Q, G339D, N343Q, R346K S371L, S373P, and S375F) have been reported in the S1 protein. Our approach was first to evaluate both trRosetta and AlphaFold algorithms with the RBD sequence of naturally occurring variants whose mutations were already previously characterized, namely the Original variant from the Wuhan, China outbreak, the Alpha variant, the Beta variant, the Gamma variant, the Delta variant, and the Omicron variant. The generated models were superimposed on the RCSB PDB:6M0J structure. The RMSD values from the AlphaFold varied greatly, so we used the trRosetta-generated models and applied the same methodology to the mutant variants [46]. The aim was to search for and identify a potential mutation with high binding affinity to the ACE-2 receptor in comparison to the known mutations.

This study was mainly focused on investigating the Receptor-Binding Motif (RBM) of the spike protein, as that portion of the protein is what specifically interacts with the ACE-2 receptor during viral infiltration and thus increases the transmissibility. The “super” command on PyMol allowed for a simple yet direct way to establish and measure the differences and accuracy of the RBD structures between different variants. After superimposing the variants, we could see some obvious differences (beta sheets disappearing, alpha helices having different amino acids on them, etc.) (Figure 1). The qualitative structural observations were performed as a preliminary way to compare the variants. The generated models were further validated using the modeling approach considering the interaction between the RBD and the ACE-2 receptor. The interaction was stabilized by several non-covalent interactions and distinctive binding stimulations. We observe a higher binding interaction of the Alpha and Omicron variants in comparison to the Original variant, suggesting that these variants could be highly transmissible. The high transmissibility of the Alpha variant resulted in the origin of the third wave of COVID-19 encountered in France [52]. However, the introduction of vaccinations coupled with acquired collective immunity to this allowed the expansion and circulation of a better-adapted variant, Delta, which eventually was monitored with vaccine introduction. Later, the reduced efficacy of vaccinations contributed to the selection of the Omicron variant, which showed a greater temporal shift [52]

The identification of P499S as a potential deleterious mutation highlights the fact that there are numerous potentially dangerous mutations that could give rise to even stronger variants than the Delta and Omicron variants causing a significant number of cases around the globe. Such mutations are not an improbability either, as the various case surges around the world present ample opportunity for various point mutations to be selected and for these variants to cause significant damage. This has already been proven by the various mutations increasing their presence in the gene pool significantly, as their docking ability may have the potential to supersede currently fixed mutations. On correlating the binding affinity, we also investigated the residue T500 and created the least likely mutagenesis to observe its effect on binding affinity. To evaluate the efficacy of the predicted mutation, we undertook a preliminary investigation of the mutation in vitro by monitoring the spike protein binding and cell-infusion assay, which correlated with the in-silico data. Although further experimental testing on P499S and T500R must be undertaken to confirm these conjectures, it would be irresponsible to ignore the potential threat of a more serious variant mutating at a time when transmission remains high across the globe.

## 5. Conclusions

In conclusion, we suggest that the higher binding affinity of P499S and T500R indicates that they could be one of the potential mutations yet to be present in nature. Omicron is a known fast-paced variant regarding transmissibility, and as per our data, these mutations show higher binding affinity in comparison to the Omicron variant. This presents additional corroboration that our computational method may be useful in determining potential future mutations. This computational model for variant investigation and de novo prediction requires further refinement. However, it still poses a significant opportunity for researchers to identify and prophylactically predict dangerous variants in relation to emerging diseases. Although the model cannot substitute lab work, it can complement research by successfully presenting a wide variety of potentially deleterious mutations and variants for broad scientific inquiry in the field.

## Figures and Tables

**Figure 1 biomolecules-12-01665-f001:**
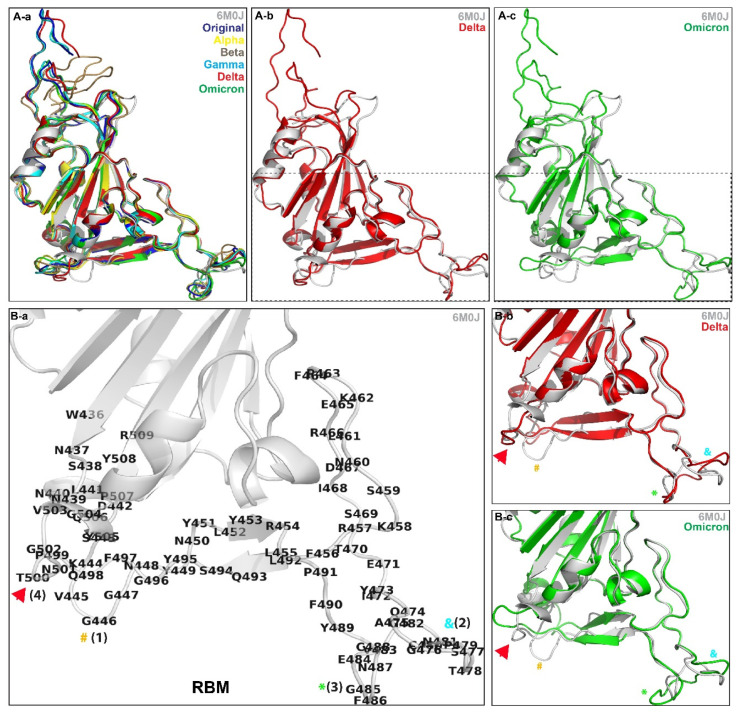
(**A-a**) All the trRosetta-generated Receptor-Binding Domain (RBD) structures were superimposed on 6M0J. (**A-b**) Structural changes between the RBM of 6M0J and the Delta sequence. (**A-c**) Structural changes of the RBM between 6M0J and the Omicron sequence. (**B-a**) RBM of 6M0J; (1), (2), (3), (4) are the changes observed. (**B-b**) Structural changes of the RBM between 6M0J and the Delta sequence. (**B-c**) Structural changes of the RBM between 6M0J and the Omicron sequence.

**Figure 2 biomolecules-12-01665-f002:**
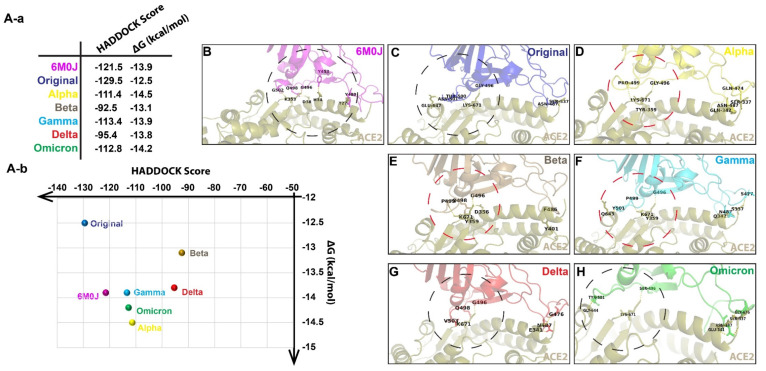
(**A-a**)The HADDOCK and PRODIGY analysis of interactions between the naturally occurring variants’ RBDs and the ACE-2 receptor from 6M0J. (**A-b**) A distribution map of the scores obtained from HADDOCK and PRODIGY. (**B**) A cartoon representation of the interaction between 6M0J RBD and the ACE-2 receptor. (**C**) A cartoon representation of the interaction between the Original RBD and the ACE-2 receptor. (**D**) A cartoon representation of the interaction between the Alpha RBD and the ACE-2 receptor. (**E**) A cartoon representation of the interaction between the Beta RBD and the ACE-2 receptor. (**F**) A cartoon representation of the interaction between the Gamma RBD and the ACE-2 receptor. (**G**) A cartoon representation of the interaction between the Delta RBD and the ACE-2 receptor. (**H**) A cartoon representation of the interaction between the Omicron RBD and the ACE-2 receptor.

**Figure 3 biomolecules-12-01665-f003:**
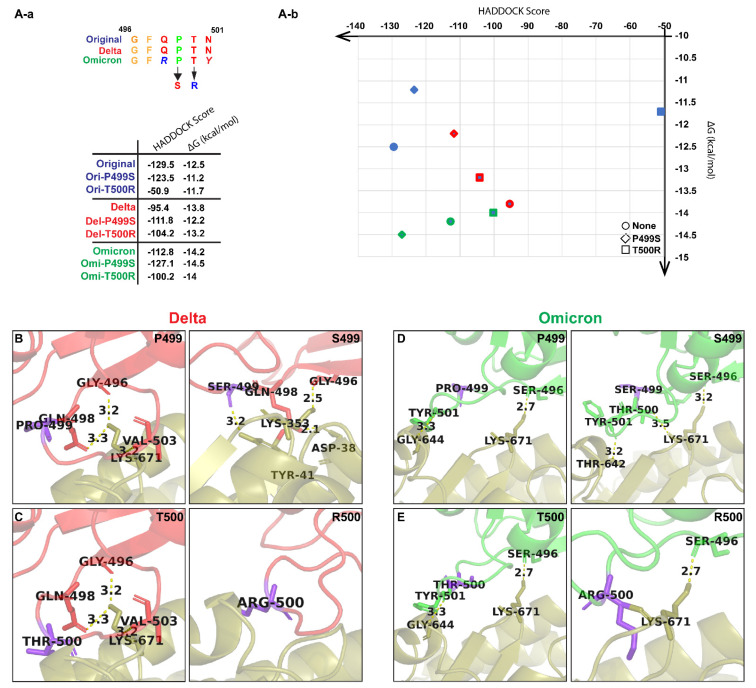
(**A-a**) Mutations in the RBM and the difference between their HADDOCK scores and binding affinity. (**A-b**) A distribution map of the scores obtained from HADDOCK and PRODIGY. (**B**) A cartoon representation of the interaction between P499 and S499 in the Delta RBD and the ACE-2 receptor. (**C**) A cartoon representation of the interaction between T500 and R500 in the Delta RBD and the ACE-2 receptor. (**D**) A cartoon representation of the interaction between P499 and S499 in the Omicron RBD and the ACE-2 receptor. (**E**) A cartoon representation of the interaction between T500 and R500 in the Omicron RBD and the ACE-2 receptor.

**Figure 4 biomolecules-12-01665-f004:**
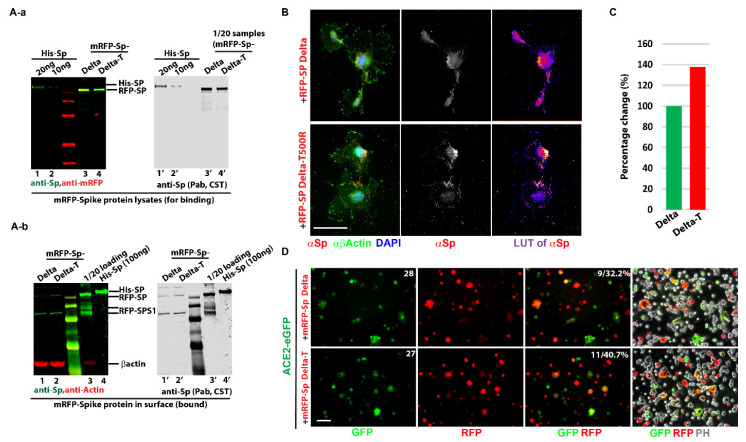
T500R increases the spike protein’s cell-bound affinity. (**A-a**) Quantifications of spike protein Delta variant (Delta) and Delta T500R (Delta-T) in transfected cell lysates (before binding). The purchased His-Sp was loaded for quantification. (**A-b**) More bound spike protein on cells was detected in T500R mutation. Spike proteins (Delta and Delta-T) were added to COS7 cells for bound, and the cell uptake level of spike proteins was checked via WB. Three experiments were repeated. (**B**) T500R mutation resulted in more endosomal entering. Delta and Delta-T were added to COS7 cells on coverslip and stained with actin (green) and spike protein (red) after a one-minute treatment with 0.0025% Trypsin to remove surface unbound ligands. Heatmap (LUT) was presented to indicate the intensity of spike protein’s stain. Warm color means high intensity. (**C**). Graph representation of the ratio of T500R/Delta (Delta-T/Delta) band’s intensity. (**D**) Cell in-fusion assay was performed to show T500R mutation enhances the ACE-2 binding affinity. Cell in-fusion was observed in the combination of ACE2-eGFP with RFP Sp-Delta or Delta-T. The efficacy was calculated as the percentage of Cells No. of “overlap of RFP with GFP”/Cells No. of “GFP”. The number was labeled inside the photos. Scale bar: 20 μm.

**Table 1 biomolecules-12-01665-t001:** RMSD (root mean square deviation of atomic positions) score for both trRosetta-generated and AlphaFold-generated RBD models superimposed on PDB: 6M0J. The root-mean-square deviation of atomic positions is the measure of the average distance (Å) between the atoms of superimposed proteins.

Models	Root-Mean-Square Deviation (RMSD) for trRosetta-Generated Models (Å)	Root-Mean-Square Deviation (RMSD) for AlphaFold v2.1.0-Generated Models (Å)
Original Variant	0.705 for 1434 atoms	1.257 for 1442 atoms
Alpha Variant	0.694 for 1446 atoms	1.596 for 1444 atoms
Beta Variant	0.777 for 1434 atoms	0.969 for 1505 atoms
Gamma Variant	0.762 for 1474 atoms	1.199 for 1488 atoms
Delta Variant	0.785 for 1466 atoms	1.407 for 1370 atoms
Omicron Variant	0.732 for 1421 atoms	0.828 for 1482 atoms

**Table 2 biomolecules-12-01665-t002:** Artificially generated variants with their residues mutated that do not naturally exist as a pathogenic variant.

Type of Variants	Residues Mutated
Variant 1	Mutations from both the Alpha and Delta variants
Variant 2	Mutation only at N501Y residue in the Original variant
Variant 3	Mutation only at S494P residue in the Original variant
Variant 4	Mutation only at S494P residue in the Delta variant
Variant 5	Mutation only at S494P residue in the Omicron variant
Variant 6	Mutation only at P499S residue in the Original variant
Variant 7	Mutation only at P499S residue in the Delta variant
Variant 8	Mutation only at P499S residue in the Omicron variant
Variant 9	Mutation only at T500R residue in the Original variant
Variant 10	Mutation only at T500R residue in the Delta variant
Variant 11	Mutation only at T500R residue in the Omicron variant

## Data Availability

Not applicable.

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
