# Peer review of "In Silico Protein Folding Prediction of COVID-19 Mutations and Variants"

_biomolecules, 2022, doi:10.3390/biom12111665_

Round 1

Reviewer 1 Report

Summary:

The manuscript entitled “In-Silico Protein Folding Prediction of COVID-19 Mutations and Variants” by Jing et al employed protein structure prediction algorithms along with molecular docking studies to understand effect of variations in mutations and its key interaction between RBM of SARS-CoV-2 and ACE-2 receptor. The authors used trRosetta, HADDOCK, and PRODIGY to model mutant structures, predict binding energy and interactions between RBM and ACE-2 receptor. Based on the in silico analysis, the authors have identified key mutations at conserved residues in SARS-CoV-2 variants. Further, the authors have performed in vitro studies to corroborate with computational data. Overall, this study is interesting but not well designed. Please find my detailed comments below.

Major comments:

1. The modeling of protein structures through de novo designing is an excellent approach to understand conformations of entirely new proteins. There are several experimentally solved structures of spike proteins deposited in public structural databases. So, I don't understand the logic behind the usage of trRosetta for modeling mutant RBM structures. Although crystal structures and models generated from ab initio methods provide a wealth of information, due to the inherent limitations of the technique, dynamic aspects such as receptor binding and subsequent structural changes are difficult to comprehend or elucidate through modeling studies. Here comes the advantage of molecular dynamics simulations. MD simulations on RBM-receptor complexes will unravel distinct structural changes adapted by the complex and provide valuable info on structural dynamics and ligand binding pockets at the interface for therapeutic intervention.

2. The authors need to elaborate on how reduced binding affinity of alpha and omicron variants increase the transmissibility? It seems the binding energy of alpha and omicron variants is highest among the docked complexes. I guess, the authors want to convey that strong binding affinity of alpha and omicron variants to ACE2 receptors increase the transmission of COVID.

3. Detailed explanation on structural changes associated with each mutant is required. For instance, how mutations alter secondary structural content, binding interface, and local interactions in the vicinity and with surrounding residues?

4. Is there any correlation between HADDOCK score and PRODIGY predicted binding energy? It seems the HADDOCK score of alpha and omicron variants is significantly less than Original and 6M0J.

5. Residue labels too small to read in figures. In fact, the authors need to adjust the size of labels and improve the quality of figures throughout the manuscript.

Reviewer 2 Report

The paper uses a protein structure prediction algorithm along with molecular docking to study the effect of variations in mutations and its key interaction between Receptor Binding Motif (RBM) of SARS-CoV-2 and Angiotensin converting enzyme 2 ACE-2 (ACE-2) receptor.

Unfortunately, there are several major problems and some fundamental concerns with the experiments. For example, as we know, AlphaFold2 is still the most accurate protein structure and even protein multimer prediction tool. Why author didn’t select AlphaFold2 to do Spike modeling and even Spike-ACE2 complex modeling? Thus, I cannot approve the manuscript in this form unless the following comments can be solved.

Major:

1.      RBD domain or spike protein has one solved structure and also has many homologies. Why not try some homologous modeling methods, like D-I-TASSER (https://zhanggroup.org/D-I-TASSER/) or AlphaFold2 (https://github.com/deepmind/alphafold).

Based on CASP results, those methods are much more accurate than trRosetta. If the authors think trRosetta is more suitable here, show the evidence, like TM-score or RMSD where trRosetta models for RBD is more accurate than AlphaFold2 etc.

2.      Also, the authors try to superpose the modeled RBD domain of Spike protein to the solved complex structure 6M0J, then use HADDOCK to refine the complex structure. Why not directly use AlphaFold2-Multimer to model the six variants of the Spike-ACE2 complex? Please show then evidence the trRosetta+HADDOCK can achieve better performance than AlphaFold2-Multimer, for example, RMSD or TM-score better.  

3.      RBD from different variants should share high sequence similarity. Thus directly using the super command in pymol in section 2.3 instead of aligning may not make sense, since the authors mentioned that they want to minimize the RMSD. The authors can try different ways to do alignment and then use RMSD to select the best method. I would suggest the authors try super, align in pymol, TMalign (https://zhanggroup.org/TM-align/), Dali (https://doi.org/10.1093/bioinformatics/16.6.566), and CE  (https://doi.org/10.1093/protein/11.9.739), and show the RMSD in the paper.

Minor:

1.    In the introduction, the authors write “in the critical assessment of protein structure prediction (CASP) experiments by Deepmind”, which is not correct. CASP is not organized by DeepMind. DeepMind team is one of the participants. This sentence is slightly misleading. I would suggest revising it as “A substantial progress, in the critical assessment of protein structure prediction (CASP) experiments, alphafold method developed by DeepMind, has been observed for improving the accuracy of protein structure prediction”

2.    In the introduction, the authors write “Protein structure prediction using Rosetta algorithm is a widely used method and 56 perhaps the most successful current method for de novo protein structure prediction since 57 its inception.” This sentence is not correct. I would suggest deleting it, benchmark results show AlphaFold2 is still the most accurate protein structure prediction program, even in the ab initio category.

3.    In the introduction, the authors write “In this study we used homology modeling, bioinformatics, and multiple-run molecular dynamics simulations to …” Why mention homology modeling here? Based on the method section, only MSA was used, and no template was used in trRosetta, thus I didn't see it as homology modeling for modeling the Spike protein RBD domain.

4.    In figure 1, red and pink are too close to make a difference. Please change to another color for the delta variant.

Reviewer 3 Report

The manuscript by Jing et al., provide important methodology and tools to predict the possible mutations in the RBD of SARS-CoV2 spike protein, which is essential to bind to the ACE receptor of host cells. Based on the earlier sequences the authors have predicted another couple of mutations which can have increased pathogenecity and deleterious effect on host.

The authors could improve the Figure 1. Figure 1 by using contrasting colors as magenta and red are hardly visible in the overlap. 

Figure 1Ba: the labelling is hardly visible.

Round 2

Reviewer 1 Report

I am happy with the revised version of the manuscript. The authors have efficiently addressed my comments and extensively modified the writeup. So, I don't have any further comments.

Author Response

Thank you for the recognition and the constructive comments over the review process.

Reviewer 2 Report

All the questions have been well addressed by the authors, I do not have further comments. 

Author Response

(The authors gave the same response as above.)
